

# Genetic structure of the small yellow croaker (*Larimichthys polyactis*) across the Yellow Sea and the East China Sea by microsatellite DNA variation: implications for the division of management units

Jian Zheng[1], Yunrong Yan[2], Zhonglu Li[2] and Na Song[1]

[1] Key Laboratory of Mariculture (Ocean University of China), Ministry of Education, Qingdao, China
[2] Guangdong Ocean University, Zhanjiang, China

## ABSTRACT

The small yellow croaker, *Larimichthys polyactis*, is a commercial fish of the order Perciformes that mainly inhabit estuaries and coastal waters. In recent years, the resources and catch of *L. polyactis* have undergone huge fluctuations. To detect genetic variations caused by the fluctuation of resources, genetic diversity of *L. polyactis* in the coastal waters of China were analyzed in this study using microsatellite DNA marker. The results revealed high genetic diversity of this species. The STRUCTURE, DAPC and $F_{ST}$ results all indicated that there was no genetic structure consistent with the distribution pattern. Overall, our main findings are in agreement with previous studies, indicating that *L. polyactis* showed high genetic diversity and low genetic differentiation. Our results for high genetic connectivity among *L. polyactis* localities provide insights into the development of management strategies, that is, to manage this species as a single management unit.

# INTRODUCTION

Marine fish have complex genetic structures because of their special geographical environment (*Hanne et al., 2010*). Relatively independent dynamic trends are usually detected in different geographic populations, yet there is also large-scale migration among them (*Gyllensten, 1985*). Therefore, genetic variation between geographical populations cannot be ignored in the assessment and management of marine fish resources. The genetic theory of adaptive evolution predicts that species rarely exist as a single breeding population due to genetic differentiation among populations (*Husband & Barrett, 1995*). Population genetic differentiation within species can generate population structure which plays an important role in evolution (*Wu, Qin & Gu, 1992*). Compared with terrestrial biota, higher genetic connectivity is usually generated in marine fishes as high migration rate and lack of physical barriers, resulting in reduced genetic differentiation (*Hewitt, 2000*).

Corresponding author
Na Song, songna624@163.com

In order to protect the genetic resources of species more reasonably, devising management strategies based on the genetic structure of the species is important. Populations with specific genetic structure are usually managed as separate conservation units (*Avise, 1992*; *Spielman et al., 2004*; *Zhang et al., 2020a*). Thus, the study of genetic diversity and genetic structure is a prerequisite for making practical conservation policy. Direct observation studies of the movements in marine organism to assess population structure in the marine environment are impractical (*Lowe & Allendorf, 2010*), and precise characterization of fish stock structure using simpler and more effective analytical approaches has long been an aim for sustainable fisheries management.

*Larimichthys polyactis* is a famous seafood product in China, and its population structure is the cause of major concern (*Zhang, Xue & Wang, 2015*). *L. polyactis* from offshore areas of China used to be divided into three populations according to the migratory route, spawning ground and morphological differences (*Zhang & Liu, 1959*; *Ikeda, 1964*; *Hu, 1998*). This division was supported by the RAPD and AFLP markers (*Meng et al., 2003*; *Han, Lin & Shui, 2009*). There are also two (*Xu & Chen, 2009*) or four management populations (*Lin, 1987*) were reported based on different methods, such as migratory route, morphological differences or fishery resources surveys (Table S1). However, mitochondrial DNA and microsatellite markers detected no significant differentiation among *L. polyactis* populations and restriction-site associated DNA sequencing verified these results (*Xiao et al., 2009*; *Kim et al., 2012*; *Li et al., 2013*; *Zhang, Xue & Wang, 2015*; *Zhang et al., 2020b*) (Table S1). The trend of younger age and miniaturization of *L. polyactis* is on the rise (*Xu & Chen, 2009*; *Lin, Liu & Jiang, 2011*; *Zhang et al., 2020a*). Therefore, more studies are needed to detect the population genetic structure of this species, thus developing more reasonable resource conservation policies.

The Yellow Sea and the East China Sea are the main habitats of *L. polyactis* where intricate hydrology and unique tectonic features have been detected in previous studies (*Lin et al., 2009*; *Ni et al., 2014*). Both freshwater outflow and ocean currents play important role in the phylogeographical patterns of *L. polyactis*. For example, the Yangtze River pours 900 billion m$^3$ freshwater into the East China Sea, which can be a barrier to block the gene flow of some marine organisms (*Su & Yuan, 2005*; *Ni et al., 2014*). Besides, the seawater with low salinity from the Yellow Sea was carries to the East China Sea by the Subei Coastal Current in summer (*Su & Yuan, 2005*). Meanwhile, the China Coastal Currents could carry the warm water from the South China Sea into the East China Sea (*Su & Yuan, 2005*), which may lead to complex habitat environment for *L. polyactis*. The generation time of *L. polyactis* is about 2 years based on previous studies (*Lin & Cheng, 2004*). However, the most recent study to focus on the genetic structure of *L. polyactis* used samples collected in 2014 (*Zhang, Xue & Wang, 2015*; *Zhang et al., 2020b*). Therefore, it is necessary to collect up to date samples to accurately study the current genetic structure of *L. polyactis* under complex geographic environments.

The complex geographic environments and the biological characteristics of migration and larval dispersal may contribute to the unique population structure of *L. polyactis* in offshore China (*Lin & Cheng, 2004*; *Lin, Liu & Jiang, 2011*). Moreover, the genetic structure of *L. polyactis* is dynamic due to the resource change caused by environment and

fishing pressure. Therefore, the current genetic structure of *L. polyactis* is the basis for the formulation and implementation of management policy. In this study, we analyzed the variation in genetic diversity and genetic structure of *L. polyactis* based on 12 polymorphic microsatellite markers developed by *Liu, Gao & Liu (2014)*. Detection of genetic diversity and current genetic structure of *L. polyactis* can help in reflecting the evolutionary potential and providing the foundation for the division of management units. Our results can be used as basis for developing more reasonable management strategies.

## MATERIAL AND METHODS

### Sample collection

A total of 168 *L. polyactis* were collected from seven localities in this study (the Yellow Sea: YT, RS, QD, LYG, YC; the East China Sea: ZS, WZ) (Table 1, Fig. 1). All fish were deposited at Fisheries Ecology Laboratory of Ocean University of China with specimen accession no. FEL202000328–FEL202000495. The geographical focus of this study was on both the Yellow Sea and the East China Sea. Previous works suggest that populations of *L. polyactis* from these areas are potentially isolated because they originate from two different overwintering grounds (*Xu & Chen, 2009*). Adult samples from the Yellow Sea belong to "the North Yellow Sea and Bohai Sea overwintering group," and samples from Zhoushan (ZS) and Wenzhou (WZ) in the East China Sea belong to "the South Yellow Sea and East China Sea overwintering group". The samples obtained in this study were approved by local fishermen, are very common, and will not cause damage to the environment. All individuals fish were identified by morphological characteristics, and then a piece of muscle of these samples was stored in 95% alcohol for total genomic DNA extraction using the phenol/chloroform method (*Sambrook, Fritsch & Maniatis, 1989*). Experiments were conducted in accordance with the 'Guidelines for Experimental Animals' of the Ministry of Science and Technology (Beijing, China; No. (2006) 398,30 September 2006).

### Primers selection and genotyping

A total of 12 microsatellite loci for *L. polyactis* developed by *Liu, Gao & Liu (2014)* were selected to study the population genetics of *L. polyactis* (Table S2). Forward primers were 5′ -labelled with a fluorescent dye (HEX, FAM or TAMRA). The PCR was performed in an A300 Fast Thermal Cycler (LongGene Scientific Instruments, Co. Ltd., Hangzhou, China). The reaction system and amplification conditions were carried out according to the methods of *Song et al. (2010)*. The annealing temperature (Ta) of each locus is showed in Table S2. The experiment was performed away from light in order to avoid fluorescence quenching. Amplicons with different fluorescent labels or different sizes were pooled and analyzed on an ABI3730 DNA sequencer (Tsingke Biotech Co., Ltd., Qingdao, China) for the genotyping of microsatellites DNA. Fragment sizes were determined with the ROX-500 standard using GeneMapper.

### Data analysis

The data was initially processed as follows. Genemarker v.1.91 was used to score the microsatellite alleles (*Hulce, Li & Snyderleiby, 2011*). The genotypes were then exported to
**Table 1** Sampling information and genetic diversity parameters of *L. polyactis* localities.

| ID | Locality | Coordinates | Sampling date | Sampling size | $H_O$ | $H_E$ | PIC | $A_R$ | $uH_E$ | $F_{IS}$ |
|---|---|---|---|---|---|---|---|---|---|---|
| YT | Yantai | 37°84′N, 121°22′E | 2019.12 | 24 | 0.976 | 0.920 | 0.889 | 11.950 | 0.917 | 0.076 |
| RS | Rushan | 36°82′N, 121°99′E | 2019.04 | 24 | 0.977 | 0.916 | 0.889 | 12.423 | 0.918 | 0.086 |
| QD | Qingdao | 35°99′N, 120°42′E | 2019.08 | 24 | 0.971 | 0.918 | 0.894 | 12.917 | 0.919 | 0.084 |
| LYG | Lianyungang | 34°52′N, 120°16′E | 2019.08 | 24 | 0.981 | 0.919 | 0.891 | 11.882 | 0.925 | 0.072 |
| YC | Yancheng | 33°78′N, 120°86′E | 2020.10 | 24 | 0.978 | 0.919 | 0.894 | 13.325 | 0.915 | 0.079 |
| ZS | Zhoushan | 29°78′N, 122°36′E | 2019.08 | 24 | 0.980 | 0.924 | 0.903 | 13.083 | 0.910 | 0.049 |
| WZ | Wenzhou | 27°84′N, 120°98′E | 2020.11 | 24 | 0.970 | 0.923 | 0.897 | 14.000 | 0.915 | 0.033 |

Notes.

$H_O$, observed heterozygosity; $H_E$, expected heterozygosity; PIC, polymorphic information content; $A_R$, allelic richness; $uH_E$, unbiased expected heterozygosity; $F_{IS}$, inbreeding coefficient.

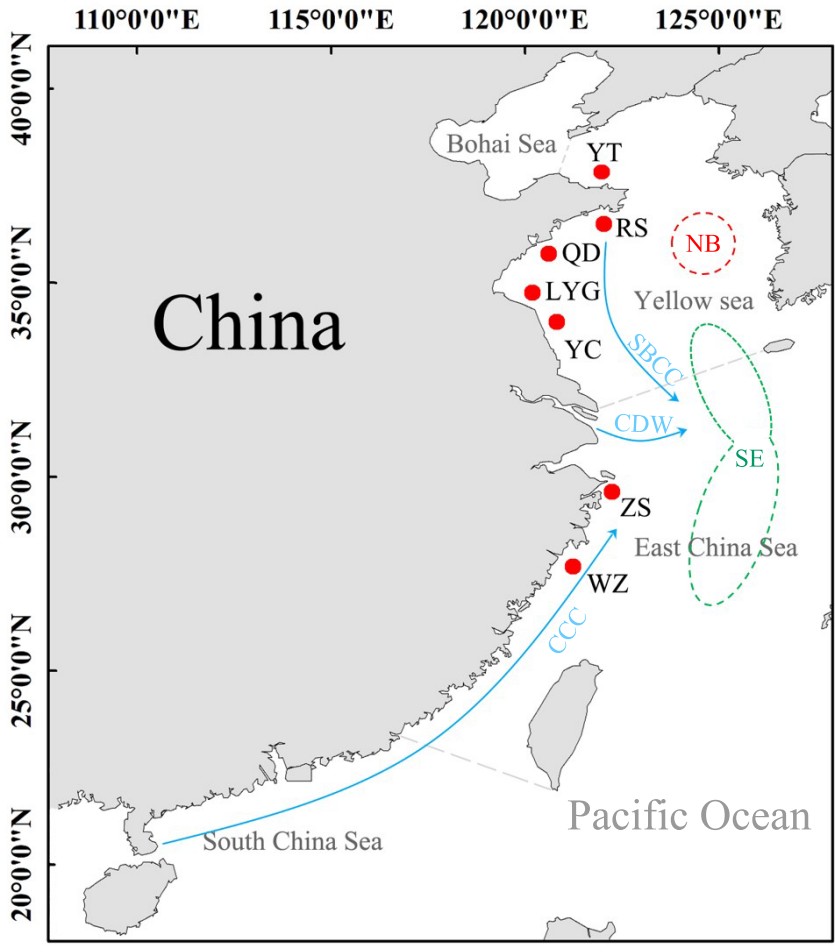

**Figure 1** **Sampling sites of *L. polyactis*.** NB: the North Yellow Sea and Bohai Sea overwintering group; SE, the South Yellow Sea and East China Sea overwintering group (*Xu & Chen, 2009*). SBCC, Subei Coastal Current; CDW, Changjiang diluted water; CCC, China Coastal Current (*Su & Yuan, 2005*).

Excel tables for data analyses. Null allele frequencies in both localities and loci using the expectation–maximization (EM) algorithm (*Dempster, Laird & Rubin, 1977*) was estimated by program FreeNA (with the number of bootstrap replicates set to 10,000) (*Chapuis & Estoup, 2007*). Using GENEPOP4.0 (*Raymond & Rousset, 1995*), the linkage disequilibrium test (*LD*) and the Hardy-Weinberg equilibrium (*HWE*) test were performed.

To analyze the variation of microsatellite loci, genetic diversity parameters at localities and locus level, such as the number of alleles ($A$), polymorphic information content ($PIC$), expected heterozygosity ($H_E$), observed heterozygosity ($H_O$) and unbiased expected heterozygosity (u $H_E$) (*Nei & Roychoudhury, 1974*), were calculated using the Excel Microsatellite Toolkit (MS-tools) (*Park, 2001*) and POPGENE 1.31 (*Park, 2001*). This software was also used to calculate allelic richness ($A_R$, a parameter of allelic number at each locus within the locality). The inbreeding coefficient ($F_{IS}$) between *L. polyactis* localities was also estimated by Fstat v.2.9 (*Goudet, 1995*).

The genetic structure was further assessed. To detect the extent of population subdivision and quantify the genetic differences, the value of $F_{ST}$ and $R_{ST}$ was obtained using FSTAT v.2.9, and the significance was evaluated using Bonferroni correction tests (*Goudet, 1995*). To test whether mutation was an important microevolutionary force acting in genetic differentiation among *L. polyactis* localities, the $F_{ST}$ and $R_{ST}$ were compared in this study (*Hardy et al., 2003*). The evidence of isolation-by-distance (IBD) was tested by regressing geographical distance and pairwise genetic distance estimate [($F_{ST}/(1- F_{ST})$), (*Rousset, 1997*) by Mantel tests using IBDWS (*Jensen, Bohonak & Kelley, 2005*). Google Earth$^{TM}$ (http://earth.google.com/) was used to measure the Euclidean distance between localities. Three-dimensional factorial correspondence analyses (3D FCA) and Discriminant Analysis of Principal Components (DAPC) were analyzed by the software of Genetix v.4.5.0. The "ggplot2 package" and "adegene package" in R 3.2.2 software was used to examine the genetic relationships among *L. polyactis* localities (*Belkhir, Borsa & Chikhi, 2004*; *R Core Team, 2015*). STRUCTURE v.2.2 was used to detect cryptic population structures which may exist within *L. polyactis* localities (*Pritchard, Tephens & Onnelly, 2000*). The parameters of the Markov chain Monte Carlo (MCMC) were set as follows: 100,000 burn-in iterations, followed by 1,000,000 iterations. The *K* value (the maximum number of clusters), estimated with the admixture model, ranged from 1–7 (total sites). To verify the results, each *K* values were run independently. To confirm the consistency of analysis, we carried out ten independent runs for each specific *K*-value. The most appropriate number of *K* value was estimated according to the change rate of the data log probability between the successive *K* values based on the *ad hoc* estimated likelihood of K (*Evanno, Regnaut & Goudet, 2005*).

To detect the evidence of recent bottleneck events, the genetic bottleneck of *L. polyactis* was estimated using the Wilcoxon's test in Bottleneck v.1.2.2 with three different mutation models: two-phased model of mutation (TPM), stepwise-mutation model (SMM) and infinite allele model (IAM) (*Piry, Luikart & Cornuet, 1999*), where 95% single-step mutations and 5% multiple step mutations with 1000 simulation iterations were set as recommended. The Mode Shift Indicator (the graphical descriptor to describe the shape

of allele frequency distribution) of Bottleneck v.1.2.2 was used to analyze allele frequency, which could differentiate between bottlenecked and stable populations.

## RESULT

### Genetic diversity of *L. polyactis*

A total of four null alleles were found in this study. The value of the null allele frequencies ranged from 0.001 (several loci) to 0.038 (ZS in locus lop116), which were all low (Table S2). There was little influence on the average genetic diversity (Table S3), so we used all loci for further analysis. One locus (Lpol05) of significant pairwise comparisons was detected based on *LD* test for localities and loci after Bonferroni correction. This locus was not kept in the further analyses. There were only four loci (Lpol05, Lpol11, Lpol15 and Lpol16) that deviated from the *HWE* after Bonferroni correction in this study.

Summary statistics of the genetic diversity parameters are shown in Table 1 and Table S4. The number of alleles ($A$) per loci ranged from nine to 20, with a total of 186 alleles ($A$) detected in seven localities. The average allele richness ($A_R$) ranged from 11.8 (WZ) to 14.0 (LYG). The average observed heterozygosity ($H_O$) of all localities ranged from 0.970 (WZ) to 0.981 (LYG) and the average expected heterozygosity ($H_E$) ranged from 0.916 (RS) to 0.924 (ZS). The range of $uH_E$ was 0.910 to 0.925. The average polymorphic information content ($PIC$), which ranged from 0.889 (YT and RS) to 0.894 (QD and YC), revealed high genetic diversity in all localities ($PIC > 0.5$). The range of $F_{IS}$ was 0.033−0.086, which these values were not significant.

### Genetic structure and differentiation

The genetic structure of *L. polyactis* localities was estimated based on pairwise $F$-statistics ($F_{ST}$). The $F_{ST}$ among localities ranged from −0.0073 (LYG *vs* ZS) to 0.0153 (QD *vs* WZ) (Table 2). Besides, the range of $R_{ST}$ was −0.0083 (RS *vs* WZ) to 0.0145 (QD *vs* ZS) (Table S5). Significant genetic differentiation was detected only between Zhoushan and a few other localities. No clear phylogeographic signal was identified due to similar value of $F_{ST}$ and $R_{ST}$, revealing low level of differentiation among *L. polyactis* localities. This result also showed that migration and genetic drift were more important relative to mutation at this sampling scale. The Mantel test indicated no significant relationship between pairwise estimates of $F_{ST}/(1- F_{ST})$ and geographic distance ($r = 0.5663$; $p = 0.142$). The result of 3D-FCA showed that the contribution rates of three principal components were 46.37%, 21.79% and 17.21%, respectively, and no significant population structure existed throughout the examined range of *L. polyactis* (Fig. S1). This result was also supported by a DAPC analysis (Fig. 2).

The cryptic population structure was detected using the software of STRUCTURE. The number of $K$ value indicated that the model where $K = 3$ was the most appropriate (Fig. 3). Seven localities did not show significant clustering trends using $K = 3$ (Fig. 4), which supported the result of $F_{ST}$ and the discriminant analysis of principal components.

**Table 2** Pairwise $F_{ST}$ (below diagonal) and pairwise geographic distances (km, above diagonal) among *L. polyactis* localities.

| Population | YT | RS | QD | LYG | YC | ZS | WZ |
|---|---|---|---|---|---|---|---|
| YT | | 86 | 230 | 426 | 696 | 1159 | 1358 |
| RS | 0.0080 | | 181 | 419 | 654 | 1137 | 1331 |
| QD | 0.0077 | 0.0012 | | 277 | 285 | 768 | 961 |
| LYG | −0.0057 | 0.0038 | 0.0049 | | 285 | 750 | 934 |
| YC | 0.0067 | −0.0076 | 0.0032 | −0.0097 | | 572 | 762 |
| ZS | 0.0002* | 0.0081 | 0.0142 | −0.0073* | 0.0115* | | 419 |
| WZ | 0.0014 | −0.0070 | 0.0153 | −0.0071 | 0.0053 | 0.0052 | |

**Notes.**
*Significant $p < 0.005$ after Bonferroni correction for multiple comparisons.

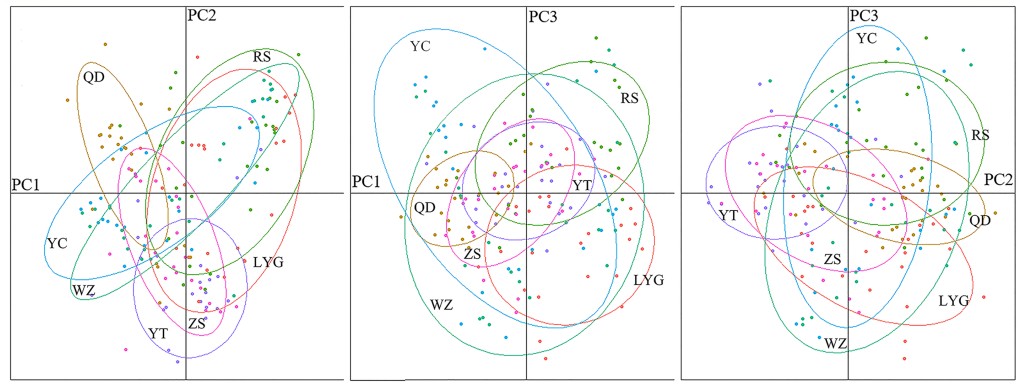

**Figure 2** Results of a discriminant analysis of principal components (DAPC).

## Detection of bottleneck effect

Under the assumption of the two-phased model of mutation (TPM), the stepwise-mutation model (SMM) and infinite allele model (IAM), seven localities did not show a recent genetic bottleneck ($P > 0.05$). Also, the L-shaped distribution of allele frequency in the mode shift test indicated that *L. polyactis* were at mutation-drift equilibrium (Table 3).

## DISCUSSION

Genetic diversity is an important component of biodiversity whose variation can generally reflect adaptive evolutionary potential (*Marty, Dieckmann & Ernande, 2015*) and disease resistance (*Zhu et al., 2013*). Polymorphic information content is an important parameter to assess the discriminatory power of molecular markers in genetic population studies (*Serrote et al., 2020*). In this study, high *PIC* values were detected, showing the usefulness of the genetic markers used in the study. The genetic diversity, measured by observed heterozygosity and expected heterozygosity, was considered high. No recently genetic bottleneck was detected, which also indicated that there was high genetic diversity in *L. polyactis*. This was consistent with the results of previous studies based on mitochondrial DNA (*Xiao et al., 2009*; *Kim et al., 2012*). The life history characters of marine organisms,

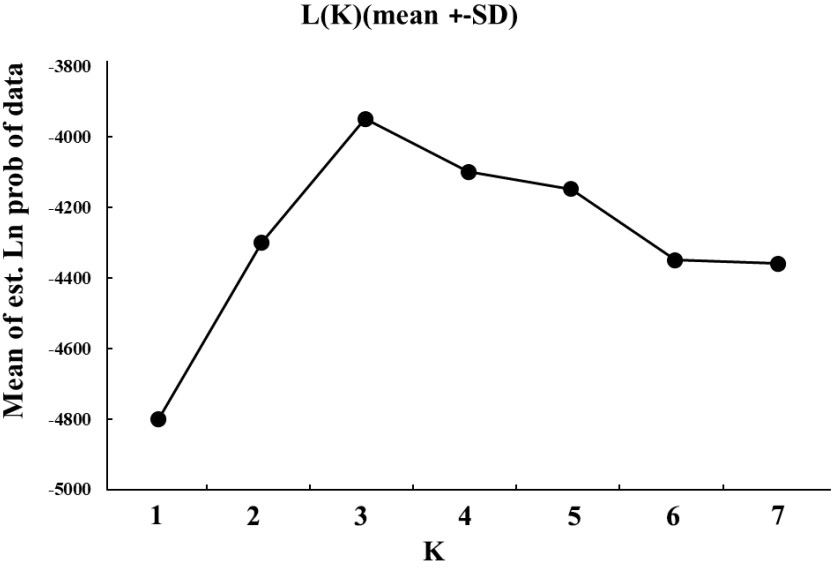

**L(K)(mean +-SD)**

**Figure 3** **The simulated K values ranged from 1 to 7 (total sites) estimated with the admixture model.**

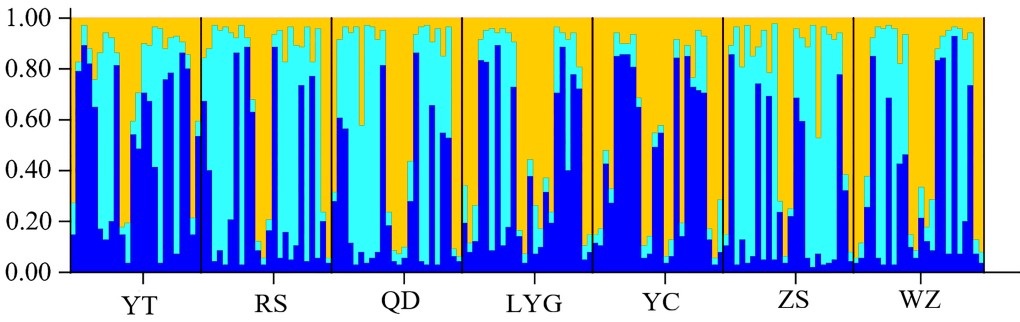

**Figure 4** **STRUCTURE bar plots from twelve microsatellite loci for seven localities of *L. polyactis*).**

such as breeding systems and larval dispersal duration, are closely related to genetic diversity (*Li et al., 2013*). Higher genetic diversity means that species may have higher evolutionary potential and stronger ability to adapt to environmental changes (*Giovannoni et al., 1990*; *Hu et al., 2021*). Both *Larimichthys crocea* and *L. polyactis* are important seafood products in China. The resources of *L. crocea* have declined rapidly because of overfishing since the 1980s, while the number of *L. polyactis* at market has remained at a high level in recent years (*Lin et al., 2009*; *Liu et al., 2016*). In the 1950s, *L. polyactis* was in the fishery boom period, whose average annual production could reach 120,0000 tons. However, in the 1960s and 1970s, it has gradually decreased due to overfishing. With the continuous increase of fishing pressure, the resource of much marine fish has declined severely in the 1980s. Some protection policies have been developed to protect the fish stock since the 1990s (*Lin, Cheng & Jiang, 2008*), which playing a key role in the recovery of *L. polyactis* resources. It is reported that the capture production of *L. polyactis* was 53,000 tons in the 1990s, while this
**Table 3** Results of Wilcoxons heterozygosity excess test, the mode shift indicator for a genetic bottleneck in *L. polyactis* localities.

| Population | Wilcoxon sign-rank test | | | Mode shift[a] |
|---|---|---|---|---|
| | IAM | TPM | SMM | |
| YT | 1.0000 | 0.8496 | 0.8203 | L |
| RS | 1.0000 | 0.8203 | 0.9101 | L |
| QD | 1.0000 | 1.0000 | 1.0000 | L |
| LYG | 0.9990 | 0.9755 | 0.9755 | L |
| YC | 1.0000 | 0.9902 | 0.9814 | L |
| ZS | 0.9755 | 0.7871 | 0.7519 | L |
| WZ | 1.0000 | 0.9980 | 0.9970 | L |

**Notes.**

Numbers in the table represent *p*-value.

[a] Normal L-shaped allele frequency distribution.

IAM, the infinite allele model; SMM, stepwise mutation model; TPM, two-phase mutation model.

data was 300,000 in the 2000s (*Li et al., 2013*). These huge fluctuations can be attributed to the strong adaptability of this species and the implementation of the summer closed fishing policy (*Cheng et al., 2004*; *Chen et al., 2020*).

As a sensitive molecular marker, microsatellite marker may detect potential genetic differentiation among localities that cannot be found based on tradition molecular markers, such as AFLP, RAPD and mitochondrial DNA (*Simbine, Viana & Hilsdorf, 2014*). It has become one of the most commonly used molecular markers in population genetics due to its characteristics of codominant inheritance and a high variability (*Gupta, Varshney & Sharma, 1999*; *Zane, Bargelloni & Patarnello, 2002*; *Parida et al., 2009*; *Askari, 2013*; *Song et al., 2017*). For instance, in contrast to works that did not find significant genetic differentiation based on other markers, *Song (2020)* detected genetic differentiation in *Acanthogobius ommaturus* between Zhoushan and other localities using 14 microsatellite loci. *Beacham et al. (2010)* also detected significant genetic differentiation in Oncorhynchus keta from Pacific Coast Honshu based on 14 microsatellite loci, which was not found by mitochondrial DNA markers and allozyme loci. In this study, no significant genetic structure consistent with the distribution pattern was detected. Only some small genetic differentiation was found between ZS and a few other localities based on the conventional population statistic $F_{ST}$. Most researchers believed that *L. polyactis* from the southern Yellow Sea and the East China Sea should be lumped together into one single group (*Lin et al., 2009*). The genetic differentiation between ZS and other localities in this study may be attributed to the complex geographical environment of the Zhoushan Islands. In the 1960s, 32° N was regarded as the geographic boundary separating the South Yellow Sea group and the East China Sea group of *L. polyactis* based on fishery-dependent studies. However, recent studies have suggested that this geographical distribution boundary might move southwards to Zhoushan Island (31° N) (*Lin et al., 2009*). *Lin et al. (2009)* also found that there was a big spawning ground of *L. polyactis* in the coastal waters of Zhoushan, which could also be the reason for the genetic differentiation. The special geographical environment of the coastal waters of Zhoushan has resulted in the complex population

structure of many species (*Song, 2020*). Also, the freshwater of Yangtze River has a wide influence on the East China Sea, including the Yangtze River Estuary fishing ground, Zhoushan fishing ground, etc (*Ni et al., 2014*; *Zhang & Hu, 2005*). Previous studies have demonstrated that the fish population near Zhoushan Island often showed different genetic characters, such as *Oplegnathus fasciatus* (*Xiao, Ma & Dai, 2016*), *Sebastiscus marmoratus* (*Xu, Song & Zhao, 2017*). Therefore, it is also vital to focus on the protection of marine organisms based on the spawning grounds and migration routes of *L. polyactis*.

There are two traditional patterns of population division in the evaluation and management of fishery resources, including geographical boundaries and administrative areas (*Ying et al., 2011*). However, these patterns are sometimes not consistent with the population structure defined by biological research, which may bring potential risks in fisheries management (*Harte, Kaczynski & Schreck, 2007*). Therefore, studying the population structure of marine fish based on different methods is important for defining management unit divisions. In this study, the results of STRUCTURE and 3D-FCA indicated no genetic structure consistent with the distribution pattern. Low or not significant $F_{ST}$ was detected in most localities, revealing high flow among *L. polyactis*. This pattern was common for other marine organisms across this area (*Zhang et al., 2020a*; *Gao et al., 2020*). *Zhang et al. (2020a)*; *Zhang et al. (2020b)* detected genetic variation in *Engraulis japonicus* in the northwestern Pacific based on restriction-site associated DNA (RAD) sequencing, suggesting high gene flow between the Bohai sea and the Yellow sea *E. japonicus* populations and no sign of local adaptation was found. High gene flow may be attributed to the spawning migrations and strong dispersal ability of *L. polyactis* (*Wirgin et al., 2000*).

Many studies have confirmed that it is important for the fishery management to consider the spatial structure of marine organisms (*Waples, 1998*; *Ying et al., 2011*). Species with significant genetic differentiation among populations should be managed separately, and if not, they are better managed jointly (*Waples, 1998*). According to the findings of the present study, *L. polyactis* in the coastal waters of China should be managed as a single management unit. There may be a single genetic stock of *L. polyactis* in different sampling sites due to long-distance migration and larval dispersal (*Hewitt, 2000*). Moreover, small genetic heterogeneity was detected between ZS and other sampling sites, which indicated that we should pay more attention. There were several separated spawning and feeding grounds across Zhoushan Archipelago sea area (*Lin et al., 2009*). Previous studies have shown that this area is the germplasm resources center of many marine organisms (*Lin et al., 2009*; *Xiao, Ma & Dai, 2016*; *Xu, Song & Zhao, 2017*). To protect marine living resources including *L. polyactis*, therefore, we suggest that the marine national nature reserve should be established in Zhoushan Archipelago sea area. Overall, effective management for important economic species should not only consider administrative division and geographical boundaries but also the biological characteristics of the species such as migration and genetic structure.

## CONCLUSION

In summary, *L. polyactis* in offshore areas of China had high genetic diversity, indicating that they had large resources and population size. No genetic structure consistent with the

distribution pattern was detected. However, a possible genetic differentiation existed in Zhoushan as detected in this study, which should arouse the attention of researchers. This study can provide the foundation for the division of management units and the formation of laws and regulations on fishery protection.

## ACKNOWLEDGEMENTS

We thank Mrs. Pengfei Li, Mr. Zhicheng Sun, Yehui Wang and Xiang Zhao for collecting the samples.

### Funding

This study was supported by the National Key R & D Program of China (2018YFD0900905). The funders had no role in study design, data collection and analysis, decision to publish, or preparation of the manuscript.

### Grant Disclosures

The following grant information was disclosed by the authors:
National Key R & D Program of China:  2018YFD0900905.

### Competing Interests

The authors declare there are no competing interests.

### Author Contributions

- Jian Zheng conceived and designed the experiments, performed the experiments, analyzed the data, prepared figures and/or tables, authored or reviewed drafts of the article, and approved the final draft.
- Yunrong Yan conceived and designed the experiments, authored or reviewed drafts of the article, and approved the final draft.
- Zhonglu Li analyzed the data, prepared figures and/or tables, and approved the final draft.
- Na Song conceived and designed the experiments, analyzed the data, prepared figures and/or tables, and approved the final draft.

### Animal Ethics

The following information was supplied relating to ethical approvals (i.e., approving body and any reference numbers):

Experiments were conducted in accordance with the 'Guidelines for Experimental Animals' of the Ministry of Science and Technology.

### Data Availability

The raw data are available in the Supplementary File.

## Supplemental Information

Supplemental information for this article can be found online at http://dx.doi.org/10.7717/peerj.13789#supplemental-information.

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
