# Peer review of "Genetic structure of the small yellow croaker (Larimichthys polyactis) across the Yellow Sea and the East China Sea by microsatellite DNA variation: implications for the division of management units"

_PeerJ, doi:10.7717/peerj.13789_

## Round 0.1 · original submission · Major Revisions

Dear authors,

I sent the manuscript out for review to determine whether the referees would identify the merits of the study that would justify publication for PeerJ. Although the reviewers recognized merit, they mention drawbacks and limitations, raising some misgivings about data analyses and the way the manuscript has been written up. They provided very constructive comments on how the manuscript can be improved. Furthermore, I included comments that should be considered. I hope that you will find all advice helpful when revising the manuscript.

Comments
(1) English is not always handled well, making some sentences difficult to understand. It needs to be gone over by a proficient speaker to clear up these problems.

(2) In the Introduction, authors should put the work in a broad context, identifying a more general/theoretical question that can be addressed with their results.

(3) The use of the word ‘population’ throughout the manuscript is not appropriate. A typical biological definition of ‘population’ is a group of interbreeding individuals that share time and space (Hedrick PW (2000) Genetics of Populations, 2nd edn. Jones and Bartlett, Sudbury, Massachusetts.), and most definitions involve some type of interbreeding. One of the goals of population genetics is to identify what is a population and how many are present. Many statistical tests were developed to identify populations. We need to be precise with our terminology and you need to demonstrate that the ‘units of sampling’ are sufficiently distinct to be considered as separate populations. It would be better if ‘sampling location’ or ‘site’ is used.

(4) Methods section (i.e., data analyses) clearly requires more effort to be attractive. In the present form, the lector has a list of results without information about the aim of these analyses. The lector doesn’t know why the authors choose these analyses; ideas must be better structured.

(5) Additional analyses such as null allele frequency, HWE test, and linkage disequilibrium test must be done as a priori analyses.

(6) Overall, the authors should improve the quality of the Tables and Figures, including the table’s titles and figure captions.

Minor comments:

Lines 77-78 – More details about the electrophoresis procedure must be given (e.g., di authors use a DNA ladder, positive and negative controls?). Please, add an image of the electrophoresis gel as supporting information.
Line 85 – Please clarify what kind of genetic distance was calculated.
Lines 109-110 – Did authors perform an IBD test to confirm this pattern?

Reviewer 1 ·

Basic reporting

The MS aims to analyze the genetic diversity present in several populations of the small yellow croakers, Larimichthys polyactis. The question is pertinent and could bring important data regarding the conservation policies since it represents one of the most famous seafood products in China and its population structure has raised big concerns. The introduction is very short but brings the most relevant data related to the main object of this work. The text contains several problems regarding the language and it must be edited by a native speaker before any new attempt of submission.

Experimental design

The most problematic part of this work is related to the methods selected. The use of Microsatellite markers to measure genetic variation is very outdated and therefore brings doubtful results. The authors based their results on only 12 markers when thousand of SNPs obtained after NGS techniques could be easily performed and provide a more robust (and trustful) amount of data to be analyzed.

Validity of the findings

As explained above, the main conclusions that " L. polyactis populations across the offshore area of
China should be protected as one management" is not fully trustful with the selected methods. Therefore, the conclusions are not fully supported by the results obtained. Nothing justifies nowadays the use of such small number of markers (microsatellites or mtDNA) in the genetic diversity of phylogenetic studies.

Reviewer 2 ·

Basic reporting

No comment.

Experimental design

More details about the Structure analysis need to be included in the manuscript, as for example: how long were the runs? What was the burnin period? How many replicates? Allelic frequency mode?

There is also a need of clarification about the genetic diversity parametrs, if they were evaluated per population, per locus or both.

Validity of the findings

No comments.

Additional comments

General aspects to review:
1) I suggest including in the introduction section information about the generation time of the species to justify why conducted the study since the last one with the same focus was published in 2015. Also, this information would make it clearer to understand that just the last generation of individuals were sampled and so just the actual genetic structure was captured by the study.
2) I recommend including information about the geographical distribution of the species and how representative are 168 individuals and seven populations to conclude that the species can be manage as only one conservation unit.
3) Please inform the readers if the markers used were developed for the species under study or for a related one.
4) In the Material and methods section please indicate if the parameters evaluated were per population, per locus or both.
5) L82: allelic abundance? What is this parameter? It is nor on the tables and it I also not discussed.
6) Please include in the material and methods section that number of alleles was also estimated.
7) Please indicate more details about the Structure analysis: how long were the runs? What was the burnin period? How many replicates? Allelic frequency mode?
8) L108-109: please review the writing of this sentence. It is possible to have genetic structure despite low values. You say that the values among all the populations except ZS were low, but were they significant? If there is one population with significant genetic structure than there is genetic structure. Now, if this genetic structure is sufficient to conclude if there is one or two units for management is other point and this must be discussed. How to establish the values from which there are one or two management units? Please if possible remove Fig 2 and present the FST values as a table.
9) Please clarify what are the 3 principal components at the DAPC analysis. Are you not only considering allelic frequencies to run the analysis. How do you explain the lack of genetic structure of population ZS in this analysis?
10) Please indicate in Fig. 4 what are the numbers below the image. I think they correspond to the sampled populations. Which one of them is ZS population? Is there any genetic structure of ZS population with the remaining populations as found in the pairwise FST analyses? Please discuss this.
11) Please include in the discussion section the genetic structure values found in the previous studies to have a better picture of the actual genetic structure and the older ones.
12) L98: please remove “were”
13) Abstract: please review as genetic variation and genetic diversity are in the same line and I think they mean the same
14) L24: replace study by studies.

Reviewer 3 ·

Basic reporting

The present paper examines population structure in a commercially important fish. I believe the study will be very useful, but the description of background information, details of methods, and discussion of results could be substantially improved.

The writing is professional and mostly clear. There are a few places where the wording is a little off, e.g. in lines 179-181 or line 82. These generally did not interfere with understanding though. There was one place that I thought was confusing:
• Line 48-49 – It is unclear what the genetic resources are that have fluctuated and led to changes in genetic structure.

The introduction provides some good background, but should be expanded. The introduction lists some previous studies in the same species, but does not include enough detail to understand exactly how these studies were performed and what they found. These details would help compare the current study to previous ones. It is clear that this species is important, but it would help readers to include more information about how the species has previously been managed and what effects management has had on species health and the fishing industry.

I have a few comments about figures and tables below:
• Figure 1 – It would help orient readers to either add some location labels on the map (e.g. major cities) or include a small box in the corner that is zoomed out so readers better understand where this study is occurring.
• Figure 2 – This figure is very effective. However, the color scale is a little confusing. It would be clearer if the authors used white as zero and then increased in darkness (red or blue) as Fst increased.
• Figure 5 – It would help readers to include sampling populations on the x axis in this figure. It is unclear what the five different groups are.

The raw microsatellite data is provided, but it would benefit from a README file that describes what it is, what programs can open it, and which files correspond to which samples/populations.

Experimental design

The research is within the aims and scope of PeerJ. The research question is clearly meaningful, and the manuscript discusses the practical implications of its findings for management. It was not clear how this study was different from previous studies in the species, particularly previous studies using microsatellites. This information would improve the reader’s understanding of the knowledge gap this study fills. I have included specific comments about the methods below:
• Line 64 – Why is QD listed as unpublished data?
• Lines 72-79 – Even though the methods of this study are similar to other studies, more details should be provided for the amplification and scoring of the microsatellites. Additionally, line 57 says these are novel microsatellites, but line 73 says they were developed in Liu et al 2014. It would also be useful to include whether the loci were developed in the same species as the current study or in a different species.
• Line 87 – Please include the R package that was used for DAPC analysis
• Lines 89-91 – Please include more details for the STRUCTURE analyses. For example, how many steps were recorded, how large was the burnin, how many repetitions were performed for each K value, other program settings like the locprior, and the programs and methods used to analyze the results and choose the best value of K.
• Additionally, why was max K set to five if there are seven sampling populations?
• Another small point is that the manuscript refers to simulated K values although nothing is really simulated – a better term might be number of clusters.
• Lines 91-93 – Similar to the STRUCTURE methods, I imagine the Bottleneck program has additional settings that would be important to include here. The results also discuss a Wilcoxon test and mode shift, which would be important to mention here in the methods.
• It was unclear what the goal of the bottleneck test was. Is there reason to believe a bottleneck has occurred in the recent past?
• The authors should consider checking the dataset for null alleles with a program like Microchecker because null alleles can affect estimates of diversity and population structure.

Validity of the findings

I agree with the overall conclusion of the study, but there are areas where substantial improvements can be made.
• Lines 107-110 – In the results the maximum Fst value is stated as 0.0142, but Figure 1 makes it look like the maximum is around 0.4, which is very high. The authors should fix this discrepancy.
• Fst between RS and QD populations is also high in Figure 1, but this relationship is never mentioned. It would be useful for the authors to discuss this result.
• Like the authors mention, marine systems usually display lower Fst values than terrestrial systems, but small values can still be biologically significant. A discussion of biological versus statistical significance would be helpful (Waples, 1998 might be particularly useful). It would also improve the manuscript to add some thoughts about the power of these microsatellites to detect significant population structure.
• Lines 159-168 – It would be helpful in the discussion to talk more about the endemic branch tribes and the evidence for them.

---

## Round 0.2 · Major Revisions

I thank the authors for their revisions which have improved the manuscript. I have received three reviews for this revised version, and all reviewers agree that the manuscript is closer to being suitable for publication in PeerJ. A number of concerns remain, however, that must be addressed. Please address all reviewer comments in a revised version.

Best Wishes,
Alison Nazareno

Reviewer 2 ·

Basic reporting

The new version of the manuscript is much improved. Almost all the suggestions and comments of the reviewers were incorporated to the revised version. However, I still dont see the difference of the present study and other conducted earlier with the same kind of molecular markers. I suggested including the generation time of the species evaluated, but the sentence included in the introduction section about this issue is confusing and I really dont get it. In addition, when was the samplig conducted? How long this species lives? One year, five?

Experimental design

The new version of the manuscript is much improved, and all the suggestions about statistical analyses were incorporated.

Validity of the findings

I agree with the discussion of the results and with the conclusions of the manuscript. I think thefindings are now valid and correspond with the analyses performed and with the main objectives of the study.

Additional comments

I have some minor comments that I would like to make:

1) L47-48: I did not understand this sentence. What is the generation time of the species? How long it lives? One year, five? Please clarify.
2) L114: remove "detected".
3) L127: change "revealed" by "revealing".
4) L128-130: this sentece should be remove to the discussion section.
5) L131: remove the second apperreance of "null alleles".
6) L132-133: How do you know that the occurrence of 4 loci with null alleles does not interfer in the results?
7) Table 4: please change the title from "7 populations" to "7 sampling locations".

Reviewer 3 ·

Basic reporting

This is my second time reviewing this manuscript, so I will be brief. Overall, I believe the study will be useful, and most of my previous comments have been addressed, but there is additional room for improvement. The language could still use a little work, but I don't think this interfered with my understanding. Please see specific comments below:

Table 1: This table is helpful, but it looks like the table legend is incorrect.
Line 43: What is meant by "based on different methods"? It would help to just describe the different methods.
Data: It doesn't look like a README has been added to the data yet. I would encourage the authors to do so.

Experimental design

The authors added a lot more detail to the methods as requested. Please see specific comments below:

Line 92: Please include a citation for MICROCHECKER. I believe this should be the correct doi: https://doi.org/10.1111/j.1471-8286.2004.00684.x
Line 99: Is UPGAMA meant to be UPGMA?
Table 4: Thank you for changing "simulated k values" in the text, but it looks like it is still in this table legend.

Validity of the findings

I agree with the overall conclusions of this study, but some improvements can be made to how the results are presented. Please see specific comments below:

Line 132: It would be useful to add some supplementary materials showing the results when loci with null alleles were removed.
Line 136: This says the highest pairwise Fst value was 0.0142, but Table 5 shows a value of 0.0153 between QD and WZ.
Table 5: I think the legend is backwards. It looks like the pairwise Fst values are below the diagonal rather than above.
Table 5: What are the (delta mu)^2 values? I don't see this mentioned anywhere else.

Reviewer 4 ·

Basic reporting

The present study tests if the Larimichthys polyactis populations are genetically structured across the Yellow and East China seas. According to the authors, the findings will be important to provide the foundation for the division of management units. This manuscript version was reviewed and improved a lot. However, I have a few points that need to be better discussed.

Overall, the manuscript is well-writing, and the language is appropriate. About the literature, I think the authors used classical papers to build the introduction and discuss the results. However, more than 90% of the references have more than five years, and this is a negative point. There are important new references that could be added in the discussion section to improve it, for example, Zhang et al. (2020 - https://doi.org/10.1016/j.fishres.2020.105505) that using SNPs to also detect high genetic connectivity in the yellow croaker populations, and Zhang et al. (2020 - https://doi.org/10.3389/fmars.2020.00374).

All figures and tables presented in this manuscript are adequate. However, in my opinion, there are a lot of tables. I recommend, to move Tables 1, 3, and 4 for the Supplementary material section. I think it will also be interesting to merge figures 3 and 4. The raw data is in accordance with the Data Sharing policy of PeerJ. However, It is not possible to know which data represent the individuals based on the ID that authors employed and not provide the meaning. I strongly recommend to the authors to make available a genotype table with all individuals, like the table used in GENEAlex.

Experimental design

The authors well-designed the experiment to test their hypothesis. The samples are well distributed in different localities considered key to identifying a genetic structure. The authors have the licenses necessary to collect the samples. I am just wondering if they have vouchers. If yes, please, it is necessary to provide the museum numbers.

The methods were described with sufficient detail; however, I have a few questions and suggestions to improve this section. Please, see the General comments.

Validity of the findings

The data is robust and shows the absence of genetic structure. It is also important to provide a genotype table because it is essential for other authors to replicate the study and used the data in future studies. I think that the discussion section will be better if the authors compare their findings with other studies that employed the focal species or other species, as I cited in the Basic reporting section.

Additional comments

General comments

Major concerns

- I was expected, based on the Abstract section, to see DAPC results, however in the Material and Methods section, the authors mention only the 3D-FCA analysis. It is important to highlight that the 3D FCA and DAPC analyses are not the same. Although both methods do not require Hardy-Weinberg Equilibrium, they are different. The three-dimension factorial correspondence analysis (3D FCA) seeks to identify genetic affinities between individuals and alleles. This analysis is performed in GENETIX. The discriminant analysis of principal components (DAPC) allows maximizing the differences between groups while minimizing variation within groups and using the package “adegenet” from R platform. Thus, the authors need to show both results. If they focus on DAPC analysis they need to detail it in the Material and Methods section and remove the 3D FCA.

- In the discussion section has a serious problem about the meaning of the PIC index. PIC measures the ability of a marker to detect polymorphisms and therefore has enormous importance in selecting markers for genetic studies. Thus, there is no association between the high PIC and high genetic diversity. Please, replace the sentence in lines 159 to 162: “Polymorphic information content is an important parameter to assess the discriminatory power of molecular markers in genetic population studies (Serrote et al., 2020). In this study, high PIC values were detected, showing the informativeness of used genetic markers. The genetic diversity, measured by Ho and He, was considered high. Besides…”
Serrote, C. M. L., Reiniger, L. R. S., Silva, K. B., dos Santos Rabaiolli, S. M., & Stefanel, C. M. (2020). Determining the Polymorphism Information Content of a molecular marker. Gene, 726, 144175.

- In the Introduction section, the authors need to bring more biological information about Larimichthys polyactis mainly about migration behavior and larva dispersal. This information is very important for the readers to understand the absence of genetic structure and it is present only in the last paragraph of the Discussion section.

Minor concerns

- Lines 21 – 23: I recommend the following sentence: “Overall, our main findings are in agreement with the previous studies, indicating…”.

- Line 26: All keywords cited by the authors are present in the title. This is inappropriate. I suggest they choose new keywords. For example "Genetic diversity", "panmictic population", and "fishery resources".

- Line 44: Please, remove “in recent years”. The references that the authors listed have at least 6 years that were published.

- The authors need to standardize the word “et al.”. According to the authors' guideline, the rule is to write “et al.”, not “et al”. Thus, please, correct the entire manuscript.

- When there are two or more references in the main manuscript, they are separated by “;”, not “,”. Please, correct this issue in the entire manuscript.

- Line 62: Should be “Material and methods”, not “Materials and methods”.

- Line 74: If the authors used muscle issues, I suppose they sacrificed the animals. If I am right, where is the voucher numbers?
- Line 99: Should be “UPGMA”, not “UPGAMA”. The authors did not show the results of genetic distance among populations. Why did you not show it?

- Line 103: Was the LOCPRIOR option (using sampling locations as prior information for clustering) used? This parameter has been shown to be informative where existing population signals are weak and is generally good to use when location information is available. See https://dx.doi.org/10.3389%2Ffgene.2013.00098?
How many times do you run each K? And was the most appropriate K obtained based on ΔK method (Evanno et al., 2005) or the K with the highest log-likelihood (Pritchard et al., 2000)?

- Line 130: It is important to add a reference after the statement “…constant or even increased slightly in the present study compared with previous data.”

-Line 132: Which molecular markers deviated from HWE? And what about the linkage disequilibrium test, did you find anything?

- Line 145: It might be a good idea to examine STRUCTURE plots for K values larger and delta K (and maybe also show them in the Supplementary Info), to determine if there is any distinct population sub-structuring.

- Line 147: Should be “Seven populations…”, not “7 population”.

- Line 174: Please, remove the letter “s” between “sites” and “because”.

- Line 175: Should be “(Hewitt, 2000)”, not “(Hewitt. 2000)”.

- Line 177: According to the Code of Zoological Nomenclature, should be “Larimichthys polyactis”, not “L. polyactis”.

- Line 179? Remove “In recent years”.

- Line 182: Remove “Relatively high”.

- Lines 192-202: It was not clear to me why the authors bring the information about endemic branch tribes if the discussion is based on the absence of genetic structure. Are there other results that support more than one genetic population?

- Line 210: Should be “FCA” instead “DAPC”.

Reference
The reference section and the in-text citations need to be reviewed. Please, use reference management and follow PeerJ’s author guidelines.
- In text-citation:
Please, look at the role to use “et al.” (not “et al”, as you did several times, for example, line 55).
According to the PeerJ’s author guidelines “Multiple references to the same item should be separated with a semicolon (;) and ordered chronologically”. This former is an issue in the entire manuscript. Please, correct it.

- Reference list
(i) References in this section are not following the author guidelines. Please, correct them.

Line 302: What is the year of this publication? In the main text (line 43) is 1987 and in the Reference list is 1985.

(ii) There are references in the list, but not in the main text. Please, provide or remove them.
Carlsson, J., McDowell, J.R., Diaz-Jaimes, P., Carlsson, J.E., Bole, S.B., Gold, J.R., Graves, J. Microsatellite and mitochondrial DNA analyses of Atlantic bluefin tuna (Thunnus thynnus) population structure in the Mediterranean Sea. Mol. Ecol. 2004, 13, 3345–3356.
Jensen, J.L.; Bohonak, A.J.; Kelley, S.T., 2005. Isolation by distance, web service. BMC Genet. 6, 13.
Pfennig, N., Trüper, H.G., 1983. Taxonomy of phototrophic green and purple bacteria: A review. Ann. Del. Past. Micro. 134(1), 9-20.
Yeh, F.C., Yang, R., Boyle, T., 1999. A Microsoft Window Based Freeware for Population Genetic Analysis. Version 1.31. University of Alberta, Canada.
Zhang, H.Y., Cheng, J.H., 2015. Geostatistical analysis on spatial patterns of small yellow croaker (Larimichthys polyactis) in the East China Sea. J. Fish. Sci. China. 12(4), 419-423.

Table and figures
Table 2
- All samples must be georeferenced. The column “Locality” is not sufficient. Please, provide the GPS for all samples.
- The authors need to explain in the legend what is “Av” means.
- Is there a difference in the sampling date that could affect the results?
Table 5
- It is important to mention in the legend the meaning of “(δμ)2”
- In the legend should be “locations”, not “locaions”.
Figure 2
- There are just five sampling sites. Where are the other two (IC and WZ) sampling sites?

---

## Round 0.3 · Minor Revisions

I thank the authors for their revisions which have improved the manuscript. However, there are some minor concerns that I have pointed out that still need to be addressed if a revised version is submitted. Please also ensure you go over the paper carefully for English. I look forward to seeing an improved draft.

(1) Authors need to clarify what method was used to estimate the frequency of null alleles. In addition, replace ‘Micro-checher’ with ‘Micro-Checker’.

(2) Why the FIS (i.e., inbreeding coefficient) was not estimated? Furthermore, when small sample sizes are used (i.e., n<50), unbiased expected heterozygosity (Nei and Roychoudhury 1974) need to be calculated.

(3) Authors need to clarify why the genetic distance estimated in Population v.1.2 was used for.

(4) Please clarify if Bonferroni correction was applied for both LD and HWE tests.

(5) Please, present the results of the frequency of null allele.

(6) I strongly recommend authors to use more appropriate methods when null alleles are identified [e.g., FreeNA - Chapuis, M.-P. and Estoup, A. (2007) Microsatellite null alleles and estimation of population differentiation. Molecular Biology and Evolution 24(3): 621-631]

(7) Table 2 - I recommend authors to replace the genetic distance with geographic distances (above diagonal). Further, clarify why the Bonferroni correction was applied to the pairwise Fst estimates.
(8) Table 3 - please add what IAM, TPM, SMM, Mode shift (L) represent. The p-values should also be presented for all estimates.

---

## Round 0.4 · Major Revisions

Although some of the points raised have been addressed, the quality of the paper still needs to be improved. I provided important concerns, as listed below, that should be taken into account when revising the manuscript. I look forward to receiving your revised manuscript.

(1) In the Introduction, authors should put the work in a broad context, identifying a more general/theoretical question that can be addressed with their results. Furthermore, based on the state-of-the-art of the species distribution and population genetic structure, It would be great if the authors should include a straightforward hypothesis.

(2) Methods section (i.e., data analyses) clearly requires more effort to be attractive. In the present form, the lector has a list of results without information about the aim of these analyses. The lector doesn’t know why the authors choose these analyses (for instance, why the genetic distance based on Raymond & Rousset 1995 was estimated?); ideas must be better structured. Other issues: LD, HWE, and null allele frequency tests should be presented as a prior genetic analyses. Null allele results should be presented as a Table; STRUCTURE requires only loci in HWE – please re-check if all loci were used in STRUCTURE analysis. More details about STRUCTURE analysis should be informed.

(3) The Discussion needs to be improved, exploring more intensively the literature. Note that the discussion is not a repetition of the results.

(4) Based in the results presented, authors need to better argue what kind of practical conservation policy must be done to ‘protect’ L. polyactis resources.

(5) Overall, the authors should improve the quality of the Tables and Figures, including the table’s titles and figure captions. Some points authors need to consider: Table 1 – please remove “Av” from all parameters; Table 2 – replace “Significant after Bonferroni correction” with “Significant p=values after Bonferroni correction for multiple comparisons”; Table 3 – please indicate what IAM, TPM, and SMM mean.

---

## Round 0.5 · Major Revisions

Dear authors,

I greatly appreciated the efforts of the authors to revise the manuscript. However, in its current form it is not suitable for publication in PeerJ. I have additional comments (please see the txt file that will be forwarded separately by PeerJ staff), all of which should be addressed if a revised version is submitted. Please ensure you go over the paper carefully for English – the text has structural and typo issues and needs to be better reframed (I strongly recommend that authors seek out a language editing service). I look forward to seeing an improved draft.

Minor comments:
Table 2 – please replace the genetic distance values with the pairwise geographic distances
Figures 1 and 3 - I do insist on an artwork improvement to these figures.

---

## Round 0.6 · Minor Revisions

Dear authors,

Although some of the points raised have been addressed, the quality of the paper still needs to be improved. I provided general comments, as listed below, that should be taken into account when revising the manuscript. Please ensure you go over the paper carefully for English. I look forward to seeing an improved draft.

Comments
Line 27 – What insights regarding management strategies were provided in the manuscript?

Line 86 – 12 or 11 polymorphic SSR markers?

Line 122 – remove ‘sequencing’

Line 130 – please replace ‘number of alleles’ with ‘number of alleles (A)’

Lines 134-136 – please remove this sentence

Lines 138-140 - Authors should test if mutation is an important microevolutionary force acting in these populations. To do that, authors need to compare FST and RST. Please see:
-Hardy, OJ, Charbonnel N, Fréville H, Heuertz M (2003) Microsatellite allele sizes: A simple test to assess their significance on genetic differentiation. Genetics, 163:1467–1482
-Hmeljevski KV et al. (2017) Do plant populations on distinct inselbergs talk to each other? A case study of genetic connectivity of a bromeliad species in an Ocbil landscape. Ecology and Evolution, 7:4704–4716

Line 141 – please replace ‘isolation’ with ‘isolation-by-distance (IBD)’

Lines 141-142 – please provide appropriate references for the ‘pairwise genetic distance’ and for the ‘IBDWS program’

Line 143 – (Eucledian) geographic distance?

Line 151-152 – please remove ‘cluster’ and ‘(Evanno et al., 2005)’

Line 170 – this result was not presented in Table S3

Lines 171-173 – this sentence needs to be better rephramed. In addition, authors need to inform what locus was removed from the analyses.

Line 180 – Hue??

Lines 189-190 – Authors need to test if there is a phylogeographic sinal (see my previous comment - lines 138-140). In addition, remove the reference listed.

Line 197 – remove ‘clusters’

Line 198/Line 202 – Authors need to double-check and standardize the nomenclature used throughout the manuscript, including those of the supplementary material: e.g. sampling locations, localites or populations? I suggest using localities throughout the manuscript.

Line 244 – please replace ‘speculated’

Tables S3, S4, and S5 need to be better presented; please revise them all carefully.

---

## Round 0.7 · Minor Revisions

Dear authors,

There are minor issues that still need to be solved before my final endorsement.

(1) Please replace '(FST/(1-FST)) by Mantel test using IBDWS (Rousset. 1997; Jensen et al., 2005)' with "[(FST/(1-FST)), Rousset 1997] by Mantel test using IBDWS (Jensen et al., 2005)'

(2) The authors need to apply a statistical test before making this claim: 'The highest average allele richness (AR) was detected in the WZ locality (14.0), while the lowest was in the LYG locality (11.8).'
Alternatively, just say: the average allele richness ranged from... to....

(3) Please replace 'This result also showed that migration was more important relative to mutation at this sampling scale (Hmeljevski et al., 2017)' with 'This result also showed that migration and/or genetic drift was more important relative to mutation at this sampling scale'

(4) Table S3 - invalid alleles or null alleles?

---

## Round 0.8 · accepted · Accept

I would like to express my appreciation to the authors for their careful review and I am happy to accept this manuscript in its current form.